

# Human pose estimation in physiotherapy fitness exercise correction using novel transfer learning approach

Aisha Naseer[1], Ali Raza[2], Hadeeqa Afzal[1], Aseel Smerat[3,4,5], Norma Latif Fitriyani[6], Yeonghyeon Gu[6] and Muhammad Syafrudin[6]

[1] Institute of Information Technology, Khwaja Fareed University of Engineering & Information Technology, Rahim Yar Khan, Pakistan
[2] Department of Software Engineering, University of Lahore, Lahore, Pakistan
[3] Centre for Research Impact & Outcome, Chitkara University Institute of Engineering and Technology, Chitkara University, Punjab, India
[4] Computer Technologies Engineering, Mazaya University College, Nasiriyah, Iraq
[5] Faculty of Educational Sciences, Al-Ahliyya Amman University, Amman, Jordan
[6] Department of Artificial Intelligence and Data Science, Sejong University, Seoul, South Korea

Corresponding authors
Yeonghyeon Gu, yhgu@sejong.ac.kr
Muhammad Syafrudin,
udin@sejong.ac.kr

## ABSTRACT

**Objective:** To introduce and evaluate an efficient neural network approach for human pose estimation and correction during physical therapy exercises using wearable sensor data.

**Methods:** We leveraged benchmark data consisting of 276,625 records from wearable inertial and magnetic sensors. A novel method termed Random Forest Long Short-Term Memory (RFL), which integrates long short-term memory and Random Forest neural networks, was implemented for transfer feature engineering. The smartphone sensor data was used to generate new temporal and probabilistic features. These features were then utilized in machine learning methods to classify physical therapy exercises. Rigorous experiments, including k-fold validation and hyperparameter optimization, were conducted to validate the performance of the RFL approach.

**Results:** The RFL approach demonstrated superior performance, achieving a remarkable 99% accuracy with the Random Forest method. The rigorous experiments confirmed the efficacy and reliability of the method in classifying physical therapy exercises.

**Conclusions:** The proposed RFL method introduces a novel feature generation approach enhancing the accuracy of physical therapy exercise classification and correction. This innovative integration not only improves rehabilitation monitoring but also paves the way for more adaptive and intelligent physiotherapy assistance systems. By leveraging sensor data and advanced machine learning techniques, it has the potential to mitigate risks associated with disabilities and major diseases, thereby offering a feasible alternative to frequent clinic visits for consistent therapist guidance.

## INTRODUCTION

Physical therapy is marked by a continually evolving theoretical and scientific foundation (*Stevens-Lapsley et al., 2023*), playing a crucial role in the rehabilitation of individuals with various disorders. Medical specialties such as cardiopulmonary medicine, neurology, orthopedics, and pediatrics can significantly benefit from the application of physical therapy, contributing to the health and well-being of individuals, families, and communities (*Yurtman & Barshan, 2014*).

Regular physical activity is vital for maintaining overall well-being, particularly in enhancing strength, flexibility, and endurance (*Asghar et al., 2023*). Physiotherapy interventions, including manual therapy, exercise programs, electrotherapy, and sensorimotor rehabilitation, cater to individuals of diverse ages and abilities. Applying exercise physiology knowledge facilitates the development of advanced exercise programs focused on strengthening the arms and legs and enhancing overall posture.

The upper limb, known as the human arm, plays a vital role in daily activities such as eating, writing, and driving, primarily focusing on object manipulation. Unfortunately, injuries and diseases can compromise its mobility and functionality. Neurological diseases such as polio, hemiplegia, paraplegia, and sclerosis can impact arm motion (*Chaparro-Rico et al., 2020*). Injuries to the lower limb are also significant, with half of all general injuries leading to activity limitations. About three-quarters of these injuries involve lower body sprains and fractures, contributing to 40% of cases of restricted bed disability. Stroke survivors may encounter challenges in regaining full function in their lower limbs (*Khemani & Hahm, 2021*). Consequently, rehabilitation therapy becomes essential to restore normal function and recover the range of motion in both the upper and lower limbs.

A crucial concern in physical therapy is the assessment of exercises and the evaluation of their effectiveness in each session. Patients and physiotherapists face psychological hurdles, including misconceptions about exercise, fear of pain, reluctance towards physical activity, mental strain associated with weight-bearing functional programs, and underestimation of capability (*Lawford et al., 2020*). Patients seek guidance from physical therapists to improve their specific weaknesses through exercises. While therapists demonstrate proper techniques, patients are responsible for performing regular exercises between visits, posing a common challenge in physical therapy (*Khemani & Hahm, 2021*).

Physiotherapy interventions are essential for individuals with physical disabilities or those aiming to regain functionality post-injury or surgery. Yet, for home-based therapy to be effective, automated assessment mechanisms are necessary to ensure the proper execution of physiotherapy exercises. Despite the availability of innovative tools and equipment that facilitate home-based physical therapy, cost-effective implementations may prove ineffective if patients perform exercises incorrectly or at an unconventional pace (*Carrera, Arequipa & Hernández, 2022*). Approaches to address these challenges include implementing incentives for exercise, fostering accountability, providing education and reassurance, customizing exercise programs, and devising monitoring and correction systems to ensure the correctness of exercises.

Prior physiotherapy exercise correction techniques (*Liao, Vakanski & Xian, 2020*), particularly those based on convolutional neural networks (CNNs) and Gaussian models, face several limitations. CNN-based approaches, while effective in capturing spatial features, struggle with temporal dependencies in sequential sensor data, making them less suitable for dynamic motion analysis. Gaussian models, on the other hand, assume a predefined statistical distribution of movement patterns, limiting their adaptability to diverse patient variations and real-world complexities. These constraints highlight the need for more flexible and robust methods to enhance physiotherapy exercise classification and correction.

## Predictive analysis with machine learning

Research conducted by *Asghar et al. (2023)* emphasized the importance of accurate and reliable observation and evaluation of arm and shoulder movements in physical therapy and fitness exercises. They developed a wrist-worn device using an MPU-6050 to capture data on a user's arm movement values. The collected data were transmitted to a microcontroller that assessed movement magnitude using an accelerometer. The actual MPU-6050 output was compared with a preset value *via* the microcontroller, and the results were processed using classification algorithms. The findings indicate that wristband data, when analyzed with machine learning techniques, accurately represent various arm and shoulder movements. The study employed four distinct machine learning algorithms for optimal accuracy: Weighted k-nearest neighbors (KNN) achieved 92% accuracy, bagged trees reached 91.6% accuracy, decision trees achieved 91.4% accuracy, and fine Gaussian support vector machine achieved 86.1% accuracy. Notably, the KNN approach demonstrated a success rate of 92% and is emerging as the most effective method. These results underscore the effectiveness of wristbands in monitoring upper-body workouts with precision.

In this study, *Hong et al. (2023)* introduced an automated method for assessing the functional movement screen (FMS) using an enhanced Gaussian mixture model (GMM). The refined GMM yielded superior scoring accuracy compared to alternative models. This methodology involved increased sampling of minority samples and manual extraction of movement characteristics from the FMS dataset, recorded using two Azure Kinect depth sensors. The GMM was trained using separate sets of feature data, each assigned a distinct score (1, 2, or 3 points). The assessment of the FMS was performed using maximum likelihood estimation. Comparative analysis revealed that the improved GMM achieved heightened scoring accuracy (0.8) compared to other models, such as traditional GMM (0.38), AdaBoost.M1 (0.7), and Naïve Bayes (0.75). Moreover, the scoring outcomes from the enhanced GMM demonstrated substantial agreement with expert scoring (kappa = 0.67). This refined Gaussian mixture approach proves effective for FMS assessment and shows potential for leveraging depth cameras in this context.

*Kanade, Sharma & Muniyandi (2023)* proposed a novel deep learning framework with an attention-guided approach for assessing movement quality (MQ). This study investigates a transformer-based architecture guided by attention for MQ assessment. To validate the proposed model, a comparative analysis was conducted against existing state-

of-the-art methods. Examining the attention maps of the model provides valuable insights into the decision-making process, thereby enhancing the interpretability of predicted assessment scores. Notably, the proposed model demonstrated significant improvements in both training and inference times, crucial for real-time systems. The attention-guided transformer-based architecture for MQ assessment exhibits promising potential for delivering swift and interpretable movement quality assessments, particularly in home-based rehabilitation contexts.

*Liu et al. (2023)* focused on expanding a wearable-based recognition system for frozen shoulder rehabilitation exercises using machine learning approaches. The proposed methods enable automatic identification of movement and silence segments, as well as accurate classification of rehabilitation exercise types. The study involved twenty subjects performing six distinct exercises, demonstrating high accuracy in recognizing movement and silence segments with a notable 95.6% accuracy and 95.83% F-score, and in classifying exercise types with a 95.58% accuracy and 95.49% F-score. These findings underscore the viability of the approach for monitoring frozen shoulder rehabilitation exercises. However, it is important to acknowledge the study's limitations, including the small sample size of twenty participants, which may not fully represent the broader population.

*Hofmann et al. (2020)* presented an activity recognition system utilizing machine learning techniques. Automating the recognition and assessment of rehabilitation exercises poses significant challenges, particularly in capturing the comprehensive nature of these activities. While basic parameters such as joint angles can be measured through various sensing modalities, the holistic recognition and assessment of entire exercises remain complex. The study aimed to examine the application of machine learning approaches and models in the context of activity recognition and assessment within rehabilitation exercises. These findings indicate that machine learning has been effectively employed for recognizing specific exercises, evaluating exercise quality, and automatically scoring exercises based on established clinical assessment scales.

*Galán-Mercant et al. (2019)* introduced a human activity recognition (HAR) system designed to monitor patients' adherence to elbow extension and flexion physiotherapy exercises in their everyday surroundings using a single wrist-worn accelerometer. The proposed approach involved a one-class classification (OCC) strategy, where training data exclusively focuses on the target class, and data representing other classes are synthetically generated from the target activity data. The study evaluated four distinct classifiers—KNN, logistic regression, support vector machine (SVM), and Naive Bayes. In intra-subject evaluation, the SVM classifier demonstrated moderate success rates of 99% for classifying the target class and 83.3% for the other classes. During inter-subject evaluation, SVM achieved success rates of 90% and 100% on a per-subject basis, establishing itself as the most effective classifier among those examined.

## Predictive analysis with deep learning

*Arrowsmith et al. (2022)* focused on developing and evaluating a system that uses a smartphone camera to automatically identify physiotherapy exercises for the lower back

and shoulders at home. This was achieved by employing a freely available pose detection system to track joint positions in video recordings of healthy individuals performing these exercises. Subsequently, a CNN was trained to classify physiotherapy exercises based on keypoint time-series data. The model's performance was evaluated across various input keypoint combinations and its robustness to changes in camera angles. The CNN model demonstrated superior performance, particularly using 12 pose estimation landmarks from the upper and lower body, achieving high accuracy in classifying low-back exercises (0.995 ± 0.009) and shoulder exercises (0.963 ± 0.020). Furthermore, assessing the robustness of keypoint detection and CNN classifiers under diverse environmental conditions is essential. Lastly, developing a smartphone application is crucial for deploying the system in remote care settings.

*Uday et al. (2022)* focused on human activity recognition using both machine learning and deep learning methodologies. The primary objective was to categorize six distinct human activities by analyzing inertial signals acquired from smartphones. Data visualization employed t-distributed Stochastic Neighborhood Embedding. Various machine learning techniques, including logistic regression, linear support vector classifier (SVC), kernel support vector machine (SVM), and decision trees, were utilized. In parallel, deep learning approaches such as long short-term memory (LSTM), bidirectional LSTM, recurrent neural network (RNN) and gated recurrent unit (GRU) were implemented using unprocessed time-series data. Performance was evaluated using metrics such as accuracy, confusion matrix, precision, and recall. Notably, among the machine learning models, linear support vector classifier, and among the deep learning models, GRU demonstrated higher effectiveness in recognizing human activities compared to other models.

*Miron & Grosan (2021)* addressed the application of machine learning in assessing the accuracy of human motion, which is more complex than gestures and action recognition. Experiments conducted on a recent dataset aimed to evaluate the efficacy of machine learning approaches in categorizing the correctness of physical rehabilitation exercises. These findings underscored the potential of machine learning approaches in this task. However, the research also highlighted a limitation where machine learning algorithms could misclassify incorrectly executed actions as correct executions of different actions.

In *Liao, Vakanski & Xian (2020)*, a novel deep learning framework was proposed to automatically evaluate the effectiveness of physical rehabilitation exercises. This framework integrates several components, including measurement tools for assessing movement quality, scoring algorithms to convert these measurements into scores, and advanced neural network models for specific movement evaluations. It introduced a novel performance measurement method using log-likelihood from a Gaussian mixture model, coupled with a deep autoencoder network for compressing data into a lower-dimensional format. The system processes the movement of individual body parts using a sophisticated spatiotemporal neural network that organizes data into temporal pyramid structures and utilizes specialized sub-networks. The effectiveness of this framework was tested on a dataset containing ten different rehabilitation exercises, marking the initial application of deep neural networks in evaluating rehabilitation exercise performance. The results

indicate that this new deep learning-based framework holds promise for assessing the quality of physical rehabilitation exercises.

The research introduced an adaptive architecture tailored for end-users to assess movements through RGB videos in *Palomares-Pecho et al. (2020)*, enabling physiotherapists to incorporate personalized exercises with minimal training instances. Real-time tracking of key body joint values from image data was achieved using deep learning-based pose estimation (*Sadeghi Bigham et al., 2023*; *Borau Bernad, Ramajo-Ballester & Armingol Moreno, 2024*) frameworks. The method underwent assessment using four physiotherapeutic exercises focused on shoulder strengthening, illustrating a reduction in physiotherapist training time and facilitating the automatic evaluation of patients' movements without continuous supervision. By leveraging deep learning-based pose estimation, the system monitors key body joints in real time from red, green and blue (RGB) videos. Physiotherapists contribute a small number of video training examples to individualize exercises for patients, mirroring the traditional rehabilitation process where therapists guide patients through demonstrative instances. The results confirmed that this approach significantly reduces physiotherapist training time and enables autonomous assessment of patients' movements, as evidenced by the evaluation of four shoulder-strengthening exercises in a physiotherapeutic context.

*Zhang, Su & He (2020)* introduced a sensor-based rehabilitation exercise recognition approach, utilizing a deep learning framework to analyze movement data recorded during rehabilitation exercises. This innovative system integrates a CNN named D-CNN and Gaussian mixture models (GMM) to identify and assess rehabilitation exercises. The D-CNN processes sensory data related to body movements during exercises, while GMMs segment input signals into diverse shapes for multiple CNN routes. The sensor CNN (S-CNN) employs an improved lossless information compression algorithm to determine the likelihood of state transitions in hidden states. The research presented test results that highlighted the distinction between the best attribute values and the test scores, utilizing collected data and various activity recognition datasets. The outcomes showcased the efficacy of the proposed SSRER system in accurately evaluating rehabilitation exercises.

The summary of previous works includes an evaluation of the utilized methods, their achieved accuracy rates, and an analysis of the limitations associated with existing approaches, as detailed in Table 1.

Therefore, this study presents an artifician intelligence (AI) driven approach to identify and correct physiotherapy exercises. To achieve this, we utilized a multi-class exercise dataset that included 10 features extracted from arm and leg movements observed during eight diverse exercises. The development of the applied deep and machine learning methods was based on this dataset. Our primary contributions to the identification and correction of physiotherapy exercises are as follows:

- We present a novel method called Random Forest long short-term memory (RFL), which integrates a LSTM network with a Random Forest method to enhance feature engineering using physical therapy exercise signal data. The initial data from physical therapy exercises is processed through the RFL method, generating newly derived

**Table 1 Analysis of physical therapy exercises detection-related literature summary.**

| Ref | Year | Research aim | Applied research techniques | Performance results |
|---|---|---|---|---|
| *Asghar et al. (2023)* | 2023 | Arm exercises monitoring. | Weighted KNN, Fine Gaussian SVM, Decision Tree and Bagged Trees. | KNN-92% |
| *Hong et al. (2023)* | 2023 | Improved GMM for FMS. | Oversampling, feature extraction, GMM training. | GMM achieved heightened scoring accuracy (0.8). |
| *Kanade, Sharma & Muniyandi (2023)* | 2023 | Attention-Guided MQA Framework | Transformer-based architecture, attention | Significant improvements in training and inference time. |
| *Liu et al. (2023)* | 2023 | Frozen shoulder exercises recognition | Wearable-based system, machine learning models | Recognizing movement/silence segments-95.6% and exercise types-95.58%. |
| *Hofmann et al. (2020)* | 2022 | ML for rehabilitation exercise assessment | Machine learning algorithms in rehabilitation | Potential for enhancing automated systems in rehabilitation. |
| *Arrowsmith et al. (2022)* | 2022 | Video-based physiotherapy monitoring | Single-camera pose detection and CNN training | For low-back (0.995 ± 0.009) and shoulder exercise classification (0.963 ± 0.020). |
| *Uday et al. (2022)* | 2021 | Human activity recognition with ML | Machine learning and deep learning techniques | Superior accuracy in human activity recognition. |
| *Miron & Grosan (2021)* | 2020 | Correctness assessment in rehab exercises | Experiments on machine learning algorithms | Potential of ML algorithms, highlighted limitations. |
| *Liao, Vakanski & Xian (2020)* | 2020 | Deep learning for rehab exercise evaluation | Gaussian mixture model deep, Spatio-temporal neural network | Promising framework for evaluating rehabilitation exercise quality. |
| *Palomares-Pecho et al. (2020)* | 2020 | Adaptive architecture for movement assessment | End-user assessment through RGB videos | Reduced physiotherapist training time and facilitated automatic evaluation. |
| *Zhang, Su & He (2020)* | 2020 | Smart Sensor-based Rehab Exercise Recognition | Deep learning framework (CNN and GMM) | Accurate evaluation of rehabilitation exercises. |
| *Galán-Mercant et al. (2019)* | 2020 | HAR system for elbow exercises | KNN, SVM, logistic regression, and Naive Bayes | LIBSVM-90%. |
| *Uslu et al. (2020)* | 2019 | Deep learning for forecasting physical activity | CNN and CAE | Significant accuracy in predicting physical functional and activity fitness levels. |

temporal and probabilistic features. These transfer features are then used as input for various machine-learning methods to improve the identification and performance of physical therapy exercises.

- We employed four advanced machine learning and deep learning networks for comparative analysis. The utilized methods include logistic regression, Gaussian Naive Bayes, decision tree, and Random Forest. Each method underwent thorough validation using k-fold and hyperparameter optimization. The Random Forest method demonstrated superior performance compared to state-of-the-art approaches, particularly when leveraging the proposed RFL feature generation.

## METHODOLOGY FRAMEWORK

This section outlines the proposed approach for identifying and correcting physical therapy exercises. We provide a comprehensive discussion of the methods employed for analysis and the resulting calculations. This section offers a detailed, step-by-step description of the proposed method.

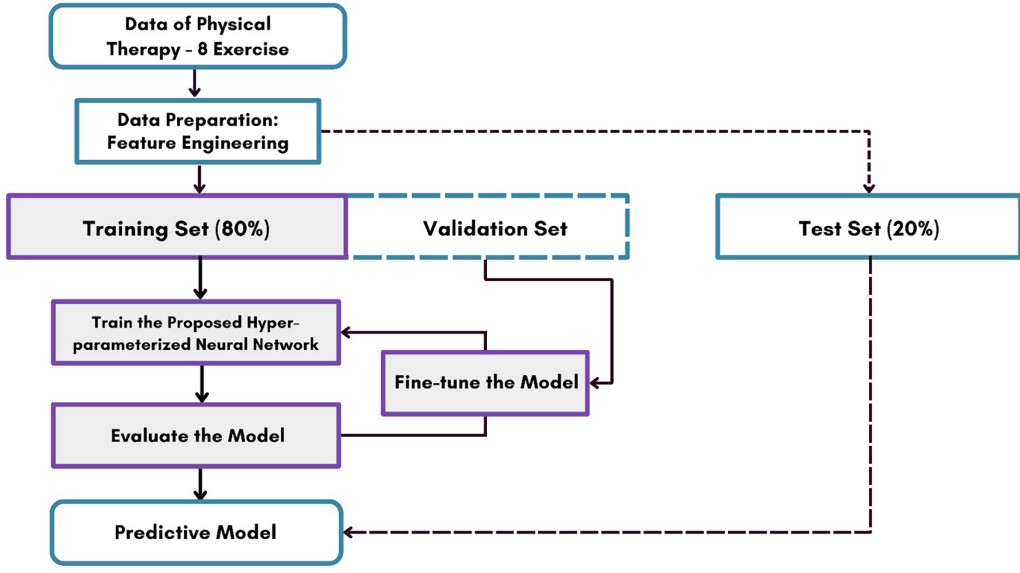

**Figure 1 The analysis of our novel proposed study methodology for classifying physical therapy exercises.**

Figure 1 illustrates the workflow architecture of the proposed methodology. Our experiment focused on eight types of arm and leg physical therapy exercises using a publicly accessible dataset. The proposed RFL method utilizes feature engineering from data collected by wearable inertial and magnetic sensors during these exercises. The resulting feature set extracted from the wearable sensor data was utilized in subsequent experiments. The dataset was divided into training (80%) and testing (20%) sets. Various advanced AI approaches were trained and tested, and the performance of hyperparameter-tuned models was evaluated using unseen test data. The approach demonstrating superior performance was then applied to identify and correct physical therapy exercises, specifically within physiotherapy fitness routines.

## Ethical considerations

This study ensures ethical compliance by utilizing publicly available sensor data, maintaining participant anonymity, and adhering to data privacy regulations. No personally identifiable information was used, and all data processing aligns with ethical guidelines for human movement analysis.

## Phase 1: physiotherapy exercise data

We utilized a benchmark dataset (*Aras & Barshan, 2022*) captured from wearable inertial and magnetic sensors during the execution of eight distinct physical therapy exercises. The experiments included multiple repetitions performed by five individuals wearing MTx sensor units produced by XSens. Each sensor unit contained three tri-axial sensors—accelerometer, gyroscope, and magnetometer—sampled at a rate of 25 Hz. The dataset properties were derived from readings of these three tri-axial sensors.

| Algorithm 1 Training and evaluation process. |
|---|
| 1: **Dataset Collection:** Collect physical therapy data with 8 different exercises. |
| 2: **Data Preparation & Feature Engineering:** Perform preprocessing, including feature selection, transformation, and normalization. |
| 3: **Splitting the Dataset:** Divide the dataset into: |
| 4: Training Set (80%)—Used to train the model. |
| 5: Validation Set—Used for hyperparameter tuning. |
| 6: Test Set (20%)—Used for final model evaluation. |
| 7: **Train the Neural Network:** Train the proposed hyper-parameterized neural network using the training set. |
| 8: **Fine-tune the Model:** Adjust model parameters based on validation set performance. |
| 9: **Evaluate the Model:** Assess model effectiveness and accuracy. |
| 10: **Develop the Predictive Model:** Finalize the trained model for predictions. |
| 11: **Test Set Evaluation:** Test the predictive model on the test set (20%) to validate performance. |

Table 2 Dataset overview—summary of dataset size, features, and train-test split.

| Attribute | Details |
|---|---|
| Total samples | 276,625 |
| Total features | 10 |
| Train samples | 221,300 (80%) |
| Test samples | 55,325 (20%) |
| Feature dimensions | 9 (excluding label) |
| Target variable | 1 (Activity Label) |

The dataset is collected from a diverse group of participants, ensuring a broad representation of demographic factors such as age, gender, physical fitness levels, and varying degrees of motor impairments. This diversity is crucial for developing a robust physiotherapy exercise correction model that generalizes well across different populations.

The dataset comprises 276,625 samples with 10 features, where nine serve as input variables, and one represents the target activity label, as shown in Table 2. It is split into 80% training data (221,300 samples) and 20% test data (55,325 samples). The target variable consists of eight distinct classes, each corresponding to a specific exercise movement. The features capture essential characteristics of these movements, aiding in accurate activity recognition. The class distribution is well-documented, with the largest category being extended leg raises (41,000 samples) and the smallest being prone lying elbow extension (31,625 samples). This structured dataset provides a solid foundation for effective model training and evaluation.

## Phase 2: preprocessing and exploratory data analysis

During the reprocessing step, we formatted the dataset and excluded the "time index" column, as it does not provide information about the performed exercise. The dataset

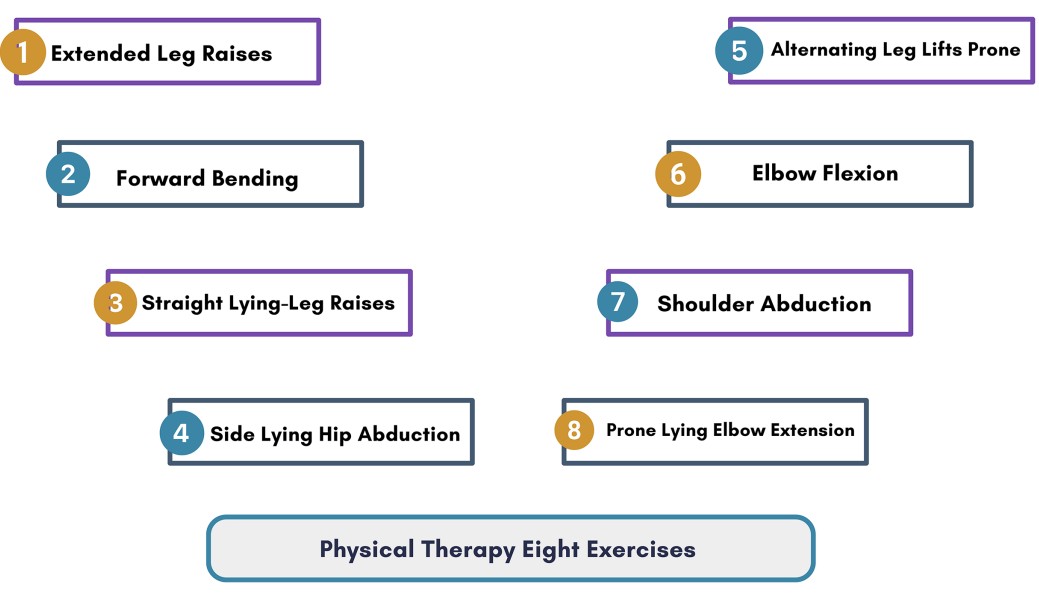

**Figure 2** The analysis of physical therapy exercise image data with their target label.

distinguishes between eight distinct target classes: Target 0, extended leg raises; Target 1, forward bending; Target 2, straight lying-leg raises; Target 3, side-lying hip abduction; Target 4, alternating leg lifts prone; Target 5, elbow flexion; Target 6, shoulder abduction; and Target 7, prone lying elbow extension, as depicted in Fig. 2. It comprises 276,625 rows and 10 columns containing attributes derived from eight types of physical therapy exercises, illustrated in Fig. 3.

## Phase 3: novel transfer feature engineering

The novel transfer-learning-based feature generation approach introduced in this study for identifying and correcting physical therapy exercises is depicted in Fig. 4. The proposed RFL method integrates LSTM and Random Forest neural networks to extract transfer features from sensor data. Initial data from wearable inertial and magnetic sensors were simultaneously processed through LSTM and Random Forest models. Subsequently, temporal and probabilistic features (*Raza et al., 2023*) were derived from LSTM and RF models, respectively. These novel transfer features served as inputs for the machine learning methods used to identify and correct physical therapy exercises in this investigation. The findings demonstrate that the proposed RFL approach achieves superior performance scores.

LSTM captures sequential dependencies in the time-series data. Given a sensor input sequence $X = \{x_1, x_2, ..., x_T\}$, the hidden state at time $t$ is computed as:

$$h_t = f(W_h h_{t-1} + W_x x_t + b_h) \tag{1}$$

where $W_h$ and $W_x$ are weight matrices, $b_h$ is the bias term, and $f$ is the activation function.

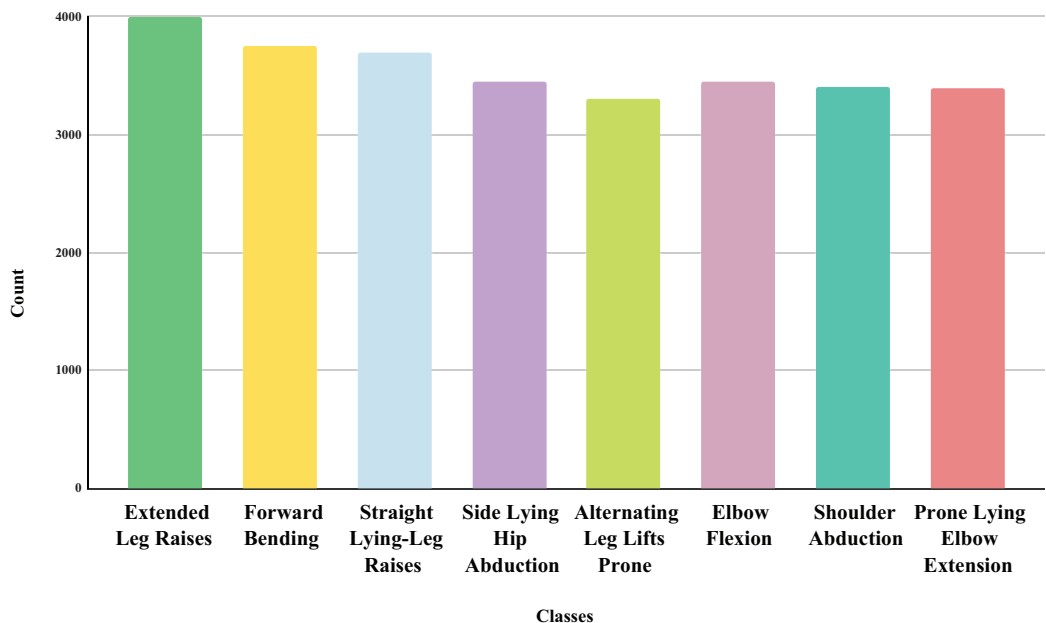

**Figure 3 The physical therapy exercise image data distribution analysis with their target label.**

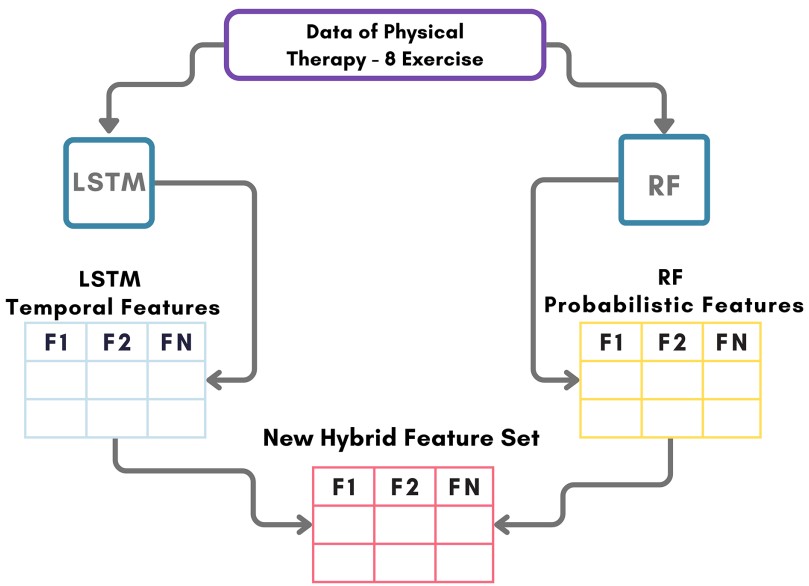

**Figure 4 Conducting a comprehensive architectural analysis of the workflow for our innovative feature engineering method introduced for the classification of images related to physical therapy exercises.**

RF generates probabilistic outputs from decision trees. Given an input feature vector $X$, the probability of class $c$ is computed as:

$$P(c|X) = \frac{1}{N} \sum_{i=1}^{N} P_i(c|X) \tag{2}$$

---

**Algorithm 2   RFL algorithm.**

**Input:** Wearable Inertial and Magnetic Sensors Features.

**Output:** Novel Transfer Learning Feature Set.

*initiate;*

1-  $RF_{tf} \longleftarrow T_{RF}(S_f)$     // here $RF_{tf}$ are the transfer features and $S_f$ are input sensor features set.

2-  $LSTM_{tf} \longleftarrow T_{LSTM}(S_f)$     // here $LSTM_{tf}$ are the transfer features and $S_f$ are input sensor features set.

3-  $F_t \longleftarrow \{RF_{tf} + LSTM_{tf}\}$     // here $F_t$ are the combined transfer feature set.

*end;*

---

where $P_i(c|X)$ is the probability estimate from the $i$-th tree and $N$ is the total number of trees.

Algorithm 2 outlines the sequential process for extracting the transfer features.

Accurately classifying physical therapy exercises is crucial for tailoring rehabilitation plans and optimizing patient outcomes and treatment effectiveness. In recent years, the adoption of machine learning techniques, particularly transfer learning-based feature engineering (*Rehman et al., 2023*), has significantly improved accuracy scores. Ensemble transfer learning (*Haider et al., 2024*), which combines multiple models to enhance prediction accuracy beyond individual capabilities, has proven effective in enhancing the accuracy of models that classify physical therapy exercises. Additionally, feature extraction involves selecting and transforming the most informative feature values from raw data, improving interpretability and generalization in these models. The synergistic use of these techniques has enabled the development of precise models for classifying physical therapy exercises, capable of accurately identifying various exercises with high recall and precision scores.

The proposed model integrates LSTM and Random Forest to extract temporal and probabilistic transfer features from sensor data. LSTM captures sequential dependencies, while RF enhances feature diversity. Machine learning models then utilize these extracted features to identify and correct physical therapy exercises, ensuring both robustness and efficiency.

## Phase 4: applied artificial intelligence approaches

Artificial intelligence, particularly deep learning techniques (*Eliwa et al., 2024*; *Mostafa et al., 2024*; *Taha et al., 2023*; *Eman et al., 2023*), has revolutionized human pose estimation in physiotherapy and fitness exercise correction. By employing advanced algorithms and neural networks, AI systems can analyze and correct body postures in real time during physiotherapy sessions or fitness exercises.

- When applied to physiotherapy fitness exercise correction, the decision tree (DT) classifier (*Wu et al., 2008*) effectively utilizes sensor data to make informed decisions. By analyzing patterns and variations in data such as movement angles, speed, and consistency, the DT classifier can identify deviations from ideal exercise execution. Assume $x_1, x_2, \ldots, x_n$ represent sensor data features such as acceleration, angular

velocity, *etc.*, used for physiotherapy fitness exercise correction. A simplified decision tree can be represented as:

$$y = \begin{cases} \text{Class 1} & \text{if } x_1 > \theta_1 \\ \begin{cases} \text{Class 2} & \text{if } x_2 \leq \theta_2 \\ \text{Class 3} & \text{otherwise} \end{cases} & \text{otherwise} \end{cases}$$

where $y$ is the output class (*e.g.*, correct or incorrect exercise form), and $\theta_1$, $\theta_2$ are thresholds determined from the training data.

- Employed in physiotherapy fitness exercise correction, Random Forest (RF) (*Ho, 1995*) effectively utilizes sensor data to enhance exercise accuracy and safety. By analyzing sensor inputs such as motion, force, and posture, this method identifies incorrect exercise forms and suggests necessary adjustments. Through its ensemble of decision trees, RF provides robust and precise feedback, significantly improving the effectiveness of physiotherapy exercises. Let $X$ represent the input sensor data vector and $Y$ the output prediction for exercise correctness. An RF method can be represented as:

$$Y = \text{RF}(X) = \frac{1}{N} \sum_{i=1}^{N} T_i(X) \tag{3}$$

where $T_i(X)$ is the result of the prediction of the $i^{th}$ decision tree in the forest, and $N$ is the total values of trees in the RF.

- Logistic regression (LR) (*Chung, 2020*) is employed in physiotherapy fitness exercise correction by analyzing sensor data to assess the correctness of exercise movements. Utilizing a binary classification system, the LR model interprets sensor inputs, such as motion or pressure data, to distinguish between correct and incorrect exercise postures. The LR model can be expressed as:

$$P(y = 1|\mathbf{x}) = \frac{1}{1 + e^{-(\beta_0 + \beta_1 x_1 + \beta_2 x_2 + \cdots + \beta_n x_n)}} \tag{4}$$

where $P(y = 1|\mathbf{x})$ is the values of probability that the exercise is performed correctly, given sensor data inputs $\mathbf{x} = (x_1, x_2, \ldots, x_n)$. Here, $\beta_0, \beta_1, \beta_2, \ldots, \beta_n$ are the parameters of the model.

- Gaussian Naive Bayes (GNB) (*Anand et al., 2022*) is effectively utilized in physiotherapy fitness exercise correction by analyzing sensor data. This probabilistic model assumes feature independence and applies Gaussian distribution to predict the likelihood of specific exercise postures. By processing sensor inputs, such as motion and force data, GNB identifies incorrect exercise forms. Given a set of features $x_1, x_2, \ldots, x_n$ obtained from physiotherapy fitness exercise sensor data, the GNB classifier estimates the posterior probability of an exercise being performed correctly (class $C$) as:

$$P(C|x_1, x_2, \ldots, x_n) = \frac{P(C) \prod_{i=1}^{n} P(x_i|C)}{P(x_1, x_2, \ldots, x_n)} \tag{5}$$

where $P(C)$ is the prior probability values of the exercise being correct, $P(x_i|C)$ is the likelihood of analyzing feature $x_i$ given that the exercise is correct, assumed to utilized a Gaussian distribution, and $P(x_1, x_2, ..., x_n)$ is the evidence, a normalizing constant. For Gaussian distributions, the likelihood $P(x_i|C)$ is given by:

$$P(x_i|C) = \frac{1}{\sqrt{2\pi\sigma_{C,i}^2}} \exp\left(-\frac{(x_i - \mu_{C,i})^2}{2\sigma_{C,i}^2}\right) \tag{6}$$

where $\mu_{C,i}$ and $\sigma_{C,i}^2$ are the mean and variance values of feature $x_i$ for class $C$.

- LSTM (*Aggarwal, 2018*) plays a pivotal role in enhancing physiotherapy fitness exercises by analyzing sensor data. These LSTM networks, adept at processing time-series data, effectively learn from sequences of movements captured by sensors, identifying patterns and deviations for accurate exercise correction. This mechanism allows for real-time feedback and tailored adjustments, significantly improving the effectiveness and safety of physiotherapy fitness routines. The layered architecture analysis of the applied LSTM is described in Table 3. The LSTM model for physiotherapy fitness exercise correction using sensor data can be formulated as follows:

$$f_t = \sigma(W_f \cdot [h_{t-1}, x_t] + b_f)$$
$$i_t = \sigma(W_i \cdot [h_{t-1}, x_t] + b_i)$$
$$\tilde{C}_t = \tanh(W_C \cdot [h_{t-1}, x_t] + b_C)$$
$$C_t = f_t * C_{t-1} + i_t * \tilde{C}_t$$
$$o_t = \sigma(W_o \cdot [h_{t-1}, x_t] + b_o)$$
$$h_t = o_t * \tanh(C_t)$$

where:

- $x_t$ is the input values sensor data at time step $t$.
- $h_t$ is the hidden state values at time step $t$.
- $C_t$ is the cell state value at time step $t$.
- $f_t$, $i_t$, and $o_t$ are the forget, input, and output gates values, respectively.
- $W$ and $b$ are the weights and bias values for different gates.
- $\sigma$ represents the sigmoid values function.
- tanh is the hyperbolic tangent values function.

- Convolutional neural network (CNN) (*Aggarwal, 2018*) is employed in physiotherapy for fitness exercise correction utilize sensor data to analyze and improve exercise execution. By processing data from accelerometers, gyroscopes, and other sensors attached to the body, the CNN effectively identifies patterns and deviations in movements, enabling precise assessment of the exercise form. This mechanism facilitates real-time feedback and correction, significantly enhancing the effectiveness and safety of physiotherapy exercises. The layered architectures of the applied CNN are described in

**Table 3 An analysis of layered architecture in applied deep learning models.**

| Layer (Type) | Output shape | Parameters |
|---|---|---|
| **CNN** | | |
| conv1d (Conv1D) | (None, 9, 8) | 32 |
| max_pooling1d (MaxPooling1 D) | (None, 2, 8) | 0 |
| flatten (Flatten) | (None, 16) | 0 |
| dropout_3 (Dropout) | (None, 16) | 0 |
| dense_3 (Dense) | (None, 8) | 16 |
| Total parameters | | |
| | 6,319,464 | |
| **LSTM** | (None, 16) | 1,152 |
| lstm_2 (LSTM) | | |
| dropout_2 (Dropout) | (None, 16) | 0 |
| dense_2 (Dense) | (None, 8) | 136 |
| Total parameters | 1,288 | |

Table 3. Given a set of sensor input data $X$, the CNN operates through a series of convolutional layers, each defined as:

$$F_l(x) = \sigma\left(B_l + \sum_{k=1}^{K} W_{lk} * x_k\right) \tag{7}$$

where $F_l(x)$ denotes the feature map in layer $l$, $\sigma$ denotes the nonlinear activation function value (*e.g.*, ReLU), $B_l$ denotes the bias, $W_{lk}$ denotes the weights of the convolution kernel, * denotes the convolution operation, and $x_k$ denotes the input from the $k$th sensor or feature map from the previous layer.

After convolutional layers, the pooling layers reduce dimensionality.

$$P_l(x) = \text{pool}(F_l(x)) \tag{8}$$

where pool is a pooling function (*e.g.*, max pooling).

The final classification layer, which is typically a fully connected layer, outputs the corrected exercise posture or classification.

$$y = \text{softmax}(W_f \cdot \text{flatten}(P_L(x)) + B_f) \tag{9}$$

where $y$ is the output vector representing different exercise postures or correction categories; $W_f$ and $B_f$ are the weights and biases of the fully connected layer, flatten transforms the feature map into a vector; and softmax provides the probability distribution over classes.

## Phase 5: hyperparameter tuning

The optimal parameters for the deep learning and machine learning approaches were determined through iterative testing and training procedures (*Aach et al., 2022*). The best-fit hyperparameters were selected using a k-fold cross-validation mechanism. Table 4

**Table 4 Analysis of the fine-tuning techniques applied in the field of artificial intelligence.**

| Techniques | Hyperparameters |
| --- | --- |
| RF | n_estimators = 10, max_depth = 10, random_state = 0, criterion = 'entropy' |
| DT | Random_state = None, max_leaf_nodes = None, min_impurity_decrease = 0.0 |
| GNB | Priors = None, var_smoothing = 1e−09 |
| LR | Random_state = 0, max_iter = 200, multi_class = 'auto', C = 1.0 |
| LSTM | Optimizer = 'adam', loss = 'categorical_crossentropy', metrics = ['accuracy'] |
| CNN | Optimizer = 'adam', metrics = ['accuracy'], loss = 'categorical_crossentropy' |

presents the optimal parameters chosen for the physical therapy exercise classification model.

## RESULTS AND DISCUSSIONS

In this section, we analyze the outcomes resulting from the application of advanced deep learning and machine learning methods. Additionally, we conducted comprehensive scientific discussions to analyze and compare the results, aiming to assess the performance of each machine learning technique employed.

### Experimental setup

In this subsection, we outline the experimental configuration used in our study. We utilized Python programming (version 3.0) to implement various machine learning techniques. The experiments were conducted in the Google Colab framework (*Carneiro et al., 2018*), leveraging a GPU backend system with 13 GB of RAM and 90 GB of disk space. All experiments related to our study were executed within this framework. To assess performance, key metrics such as precision score, accuracy score, recall score, and F1-scores were employed.

### Results with original sensors features

Table 5 provides a detailed analysis of these key performance metrics, including F1-score, precision, recall, and accuracy. These metrics are crucial in machine learning and were evaluated using sophisticated methods applied to the features of the original dataset. The analysis of results indicated that the RF, DT, GNB, and LR machine learning techniques exhibited varying performances compared to RF. Although RF achieved a commendable score of 0.96, it was not the highest score obtained, suggesting the potential for further enhancements to optimize the classification of physical therapy exercises. Moreover, the performance metrics analysis revealed that RF outperformed other techniques in terms of accuracy, recall, and precision. The notable recall rate of the RF model indicates its effectiveness in detecting a significant number of positive instances. In conclusion, improving these performance scores is crucial for achieving optimal classification accuracy in identifying physical therapy exercises.

The objective of this research is to evaluate the effectiveness of various machine learning approaches based on key metrics such as recall, F1-score, precision, and accuracy, as

**Table 5 Analysis results of performance metrics with original features using machine learning models.**

| Technique | Accuracy | Target exercise | Precision | Recall | F1 | Support |
|---|---|---|---|---|---|---|
| RF | 0.96 | Extended leg raises | 0.91 | 0.94 | 0.93 | 7,951 |
| | | Forward bending | 0.99 | 0.93 | 0.96 | 7,448 |
| | | Straight lying-leg raises | 0.94 | 0.96 | 0.95 | 7,386 |
| | | Side lying hip abduction | 0.98 | 0.99 | 0.98 | 6,484 |
| | | Alternating leg lifts prone | 0.93 | 0.96 | 0.94 | 6,273 |
| | | Elbow flexion | 0.96 | 0.99 | 0.98 | 6,711 |
| | | Shoulder abduction | 0.98 | 0.91 | 0.94 | 6,692 |
| | | Prone lying elbow extension | 1.00 | 1.00 | 1.00 | 6,380 |
| | | Average | 0.96 | 0.96 | 0.96 | 55,325 |
| DT | 0.88 | Extended leg raises | 0.88 | 0.84 | 0.86 | 7,951 |
| | | Forward bending | 0.85 | 0.83 | 0.84 | 7,448 |
| | | Straight lying-leg raises | 0.87 | 0.90 | 0.88 | 7,386 |
| | | Side lying hip abduction | 0.86 | 0.97 | 0.91 | 6,484 |
| | | Alternating leg lifts prone | 0.91 | 0.87 | 0.89 | 6,273 |
| | | Elbow flexion | 0.77 | 0.95 | 0.85 | 6,711 |
| | | Shoulder abduction | 0.94 | 0.77 | 0.85 | 6,692 |
| | | Prone lying elbow extension | 0.99 | 0.93 | 0.96 | 6,380 |
| | | Average | 0.88 | 0.88 | 0.88 | 55,325 |
| GNB | 0.40 | Extended leg raises | 0.41 | 0.25 | 0.31 | 7,951 |
| | | Forward bending | 0.28 | 0.69 | 0.40 | 7,448 |
| | | Straight lying-leg raises | 0.52 | 0.49 | 0.50 | 7,386 |
| | | Side lying hip abduction | 0.46 | 0.60 | 0.52 | 6,484 |
| | | Alternating leg lifts prone | 0.39 | 0.23 | 0.29 | 6,273 |
| | | Elbow flexion | 0.45 | 0.54 | 0.49 | 6,711 |
| | | Shoulder abduction | 0.32 | 0.13 | 0.19 | 6,692 |
| | | Prone lying elbow extension | 0.52 | 0.20 | 0.29 | 6,380 |
| | | Average | 0.42 | 0.40 | 0.38 | 55,325 |
| LR | 0.32 | Extended leg raises | 0.39 | 0.60 | 0.39 | 7,951 |
| | | Forward bending | 0.47 | 0.36 | 0.47 | 7,448 |
| | | Straight lying-leg raises | 0.29 | 0.33 | 0.29 | 7,386 |
| | | Side lying hip abduction | 0.04 | 0.02 | 0.03 | 6,484 |
| | | Alternating leg lifts prone | 0.16 | 0.12 | 0.14 | 6,273 |
| | | Elbow flexion | 0.31 | 0.46 | 0.37 | 6,711 |
| | | Shoulder abduction | 0.08 | 0.05 | 0.06 | 6,692 |
| | | Prone lying elbow extension | 0.48 | 0.53 | 0.51 | 6,380 |
| | | Average | 0.28 | 0.32 | 0.29 | 55,325 |

depicted in Fig. 5. The comparison, illustrated in a bar chart, reveals that the LR, DT, and GNB models exhibited subpar performance across all metrics. In contrast, the RF approach demonstrated highly favorable outcomes, scoring an impressive 96%. It is

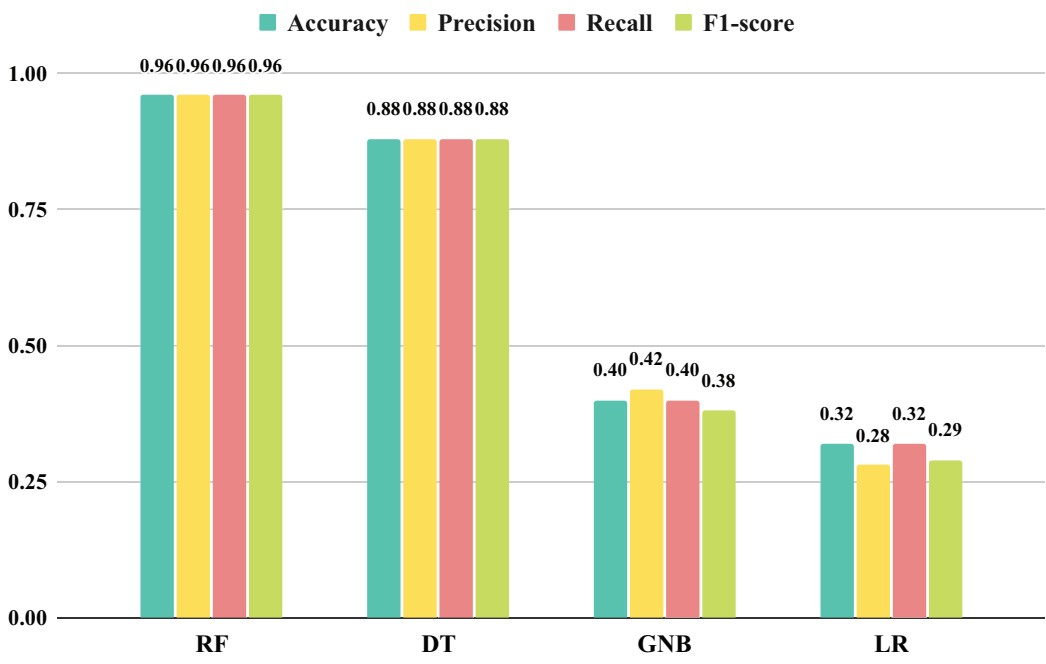

**Figure 5 The histogram-based performance outcomes analysis of applied machine learning models with original features.**

important to note that the features utilized in this analysis did not yield satisfactory results, emphasizing the necessity for additional feature extraction or selection. These findings underscore the importance of careful model selection and feature engineering to achieve optimal performance in machine learning.

The evaluation of the performance of the employed deep learning methods in a time series context is illustrated in Fig. 6. This analysis involves assessing the performance metrics throughout the training of LSTM and CNN models. Over ten training epochs, there was a notable increase in loss scores and a simultaneous decrease in accuracy scores. The examination reveals that both LSTM and CNN, as deep learning-based methods, yielded suboptimal performances on the given dataset. These observations suggest that LSTM and CNN methods did not achieve satisfactory scores in classifying physical therapy exercises, as illustrated in Fig. 7.

The assessment of performance results for unseen testing data using deep learning-based LSTM and CNN is presented in Table 6. The examination of the unseen testing data reveals that the LSTM model achieved an accuracy score of 0.88, whereas the CNN model showed an accuracy score of 0.47 in classifying physical therapy exercises. This analysis leads to the conclusion that the employed deep learning models exhibited subpar performance in exercise classification.

The analysis depicted in Fig. 8 discusses the outcomes of a comparative radar chart analysis (*Chen et al., 2022*) applied to machine-learning approaches based on the original features. Radar charts proved to be a robust method for representing the performance of each model, with each point on the chart illustrating the strengths and weaknesses of the
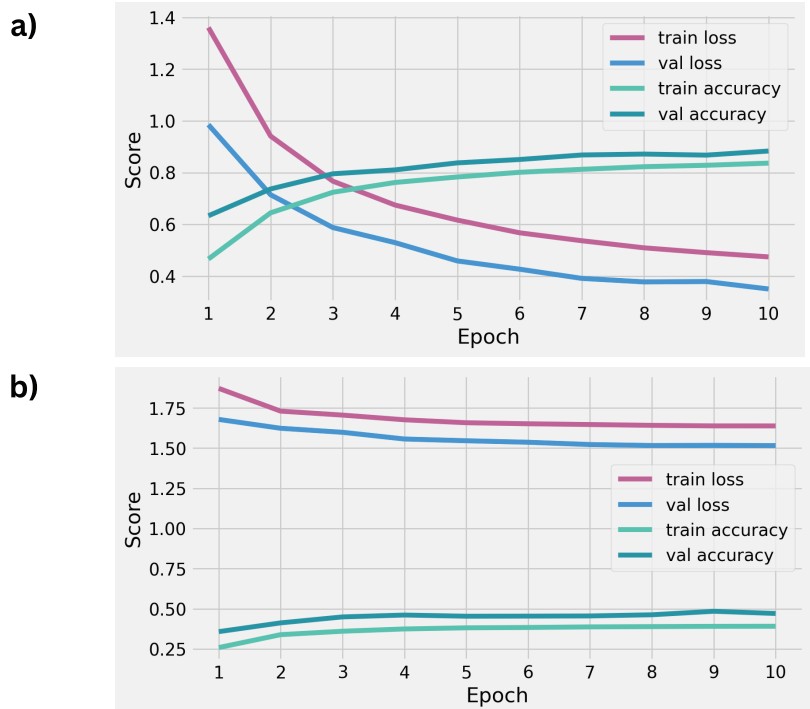

**Figure 6** The results analysis of performance for applied LSTM (A) and CNN (B) deep learning models with original features.

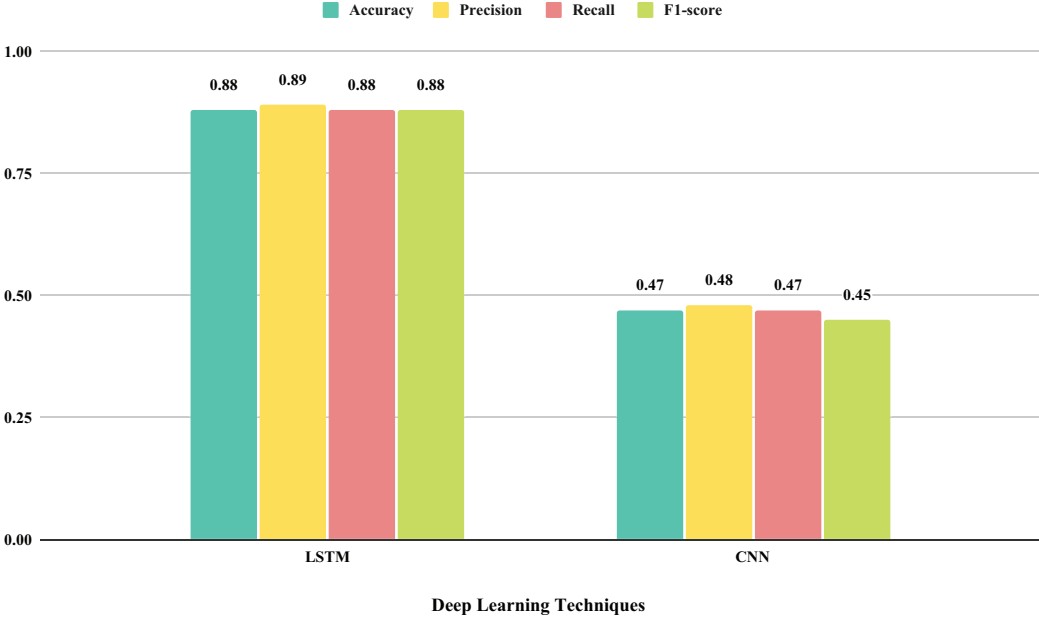

**Figure 7** An analysis of performance outcomes using histograms is conducted on deep learning models that utilized original features.

**Table 6 Analysis results of performance metrics with original features with implemented deep learning models.**

| Technique | Accuracy | Target exercise | Precision | Recall | F1 | Support |
|---|---|---|---|---|---|---|
| LSTM | 0.88 | Extended leg raises | 0.84 | 0.89 | 0.86 | 7,951 |
| | | Forward bending | 0.90 | 0.90 | 0.90 | 7,448 |
| | | Straight lying-leg raises | 0.86 | 0.92 | 0.89 | 7,386 |
| | | Side lying hip abduction | 0.93 | 0.93 | 0.93 | 6,484 |
| | | Alternating leg lifts prone | 0.86 | 0.83 | 0.84 | 6,273 |
| | | Elbow flexion | 0.89 | 0.94 | 0.91 | 6,711 |
| | | Shoulder abduction | 0.88 | 0.78 | 0.82 | 6,692 |
| | | Prone lying elbow extension | 0.94 | 0.88 | 0.91 | 6,380 |
| | | Average | 0.89 | 0.88 | 0.88 | 55,325 |
| CNN | 0.47 | Extended leg raises | 0.40 | 0.49 | 0.44 | 7,951 |
| | | Forward bending | 0.43 | 0.47 | 0.45 | 7,448 |
| | | Straight lying-leg raises | 0.49 | 0.62 | 0.55 | 7,386 |
| | | Side lying hip abduction | 0.50 | 0.67 | 0.57 | 6,484 |
| | | Alternating leg lifts prone | 0.55 | 0.50 | 0.52 | 6,273 |
| | | Elbow flexion | 0.44 | 0.55 | 0.49 | 6,711 |
| | | Shoulder abduction | 0.41 | 0.12 | 0.19 | 6,692 |
| | | Prone lying elbow extension | 0.62 | 0.32 | 0.42 | 6,380 |
| | | Average | 0.48 | 0.47 | 0.45 | 55,325 |

models across various aspects. The visual representation highlights that the RF technique demonstrated superior performance by covering a broader area under the radar span across various performance metrics curves. The findings indicate that DT, GNB, and LR exhibited lower performance scores in the radar chart analysis.

A comprehensive examination of model performance is presented in Fig. 9 through a detailed analysis of the confusion matrix. This assessment evaluates and summarizes the effectiveness of the machine learning approaches used in this study. The results indicate that the DT, GNB, and LR methods exhibited elevated error rates in target class prediction when utilizing the original features, suggesting suboptimal results for accurately categorizing the data. In contrast, the RF method demonstrates significantly reduced error rates, as evidenced by the corresponding confusion matrix. These observations highlight the critical importance of judiciously selecting feature extraction and classification methods to achieve optimal results in machine learning applications. Furthermore, the findings underscore the potential for improved model performance through the incorporation of ensemble learning-based feature engineering specific to this dataset.

A thorough examination of the confusion matrix is presented in Fig. 10 to evaluate and succinctly depict the effectiveness of the deep learning approaches employed in this investigation. The assessment revealed that when utilizing the original features, the CNN technique exhibited elevated error rates for the target classes, indicating suboptimal accuracy in data classification. In contrast, the LSTM method demonstrated significantly reduced error rates for the target classes, as evidenced by the LSTM confusion matrix. To

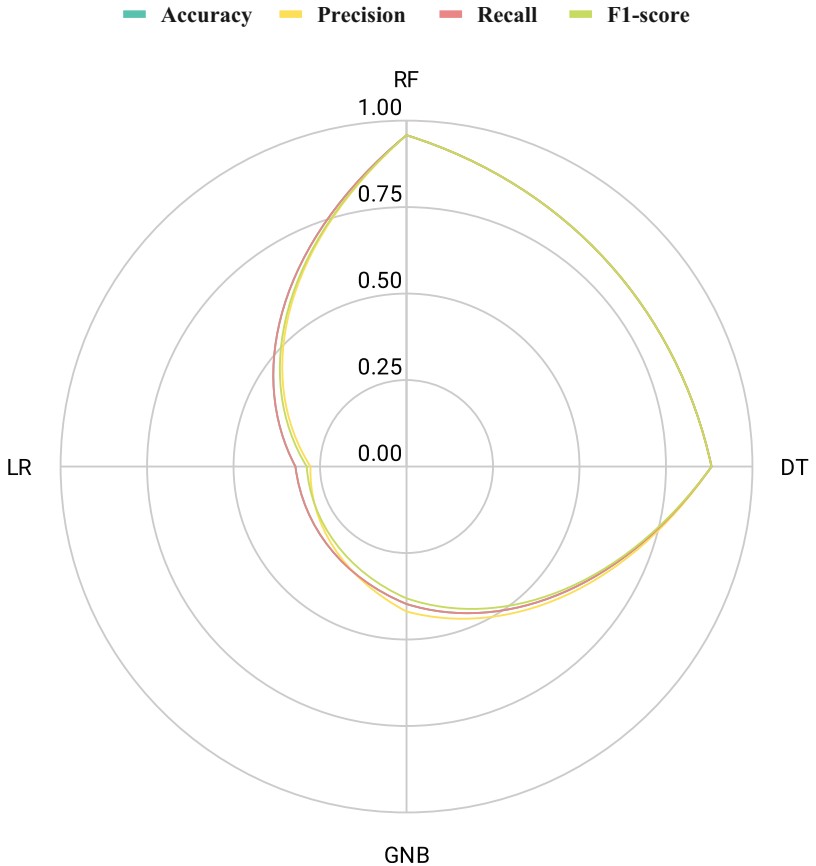

**Figure 8** An assessment of the performance results achieved by employed methods utilizing original features through radar chart analysis.

further enhance the performance of deep learning on this sensor dataset, incorporating ensemble learning-based feature generation is essential.

## Results with novel proposed transfer features

Our research aims to evaluate the effectiveness of different methods in correcting physical therapy exercise tasks through an innovative approach to feature engineering. Table 7 presents the results of the applied approaches with novel feature extraction. Notably, these methods showed significant improvements in accuracy scores with the introduction of our proposed feature generation approach. The DT, GNB, and LR approaches achieved accuracy scores of 0.98, 0.97, and 0.96, respectively. Remarkably, the proposed RF method emerged as the top-performing approach with an accuracy of 0.99, highlighting its superiority over the other methods employed. These findings strongly indicate that our feature extraction approach successfully enhances the performance of all tested methods across various metrics. This analysis underscores the importance of integrating effective feature extraction techniques to optimize machine learning approaches for classifying physical therapy exercises.

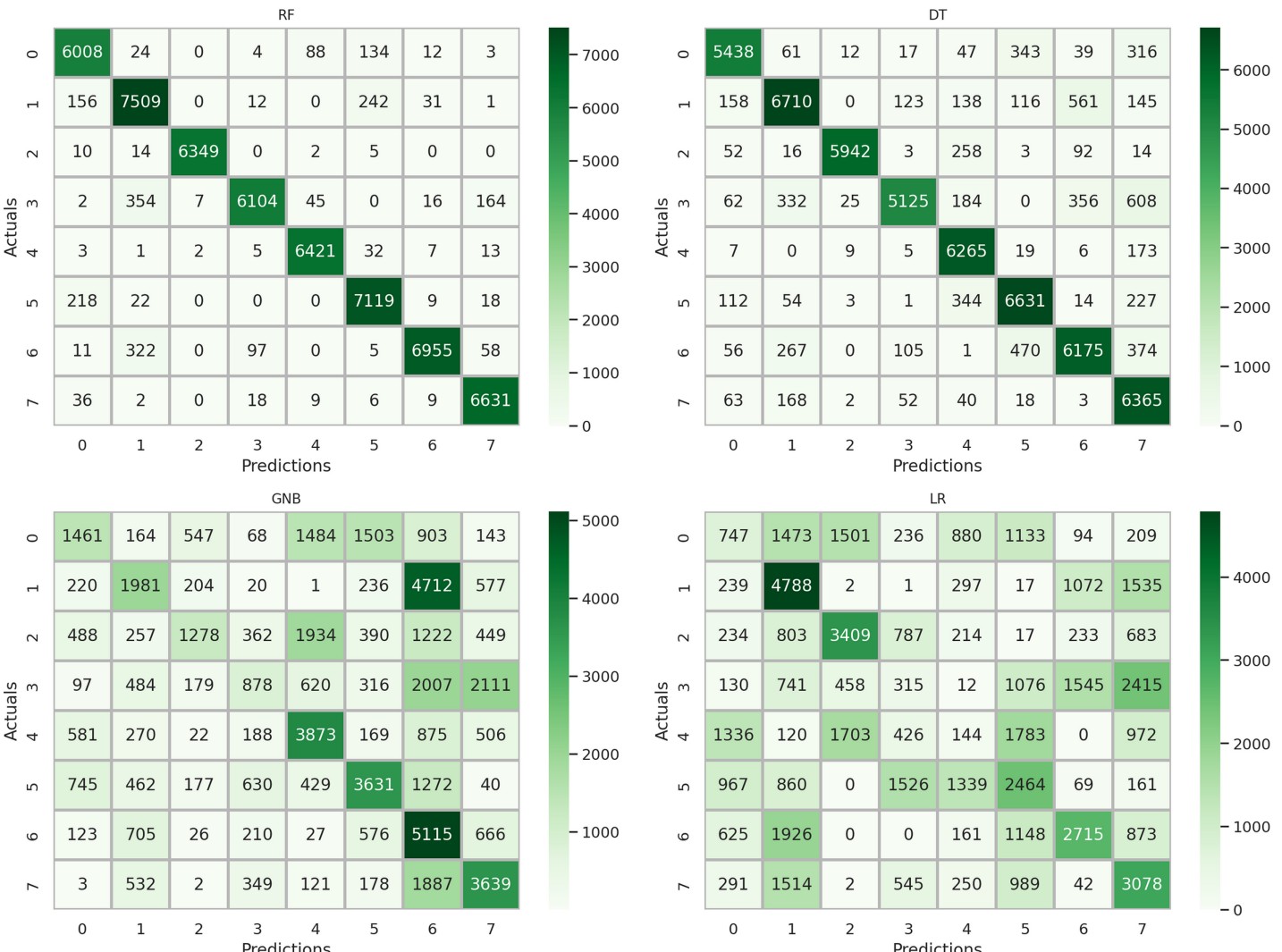

**Figure 9 Analysis of the confusion matrix outcomes for the employed machine learning methodologies using the original features.**

The evaluation of different machine learning models was conducted through a histogram-based bar chart analysis, assessing recall, F1-score, precision, and accuracy results, as depicted in Fig. 11. The comparison presented in the bar chart indicates that the DT, GNB, and LR approaches demonstrated commendable performances across all metrics, although not the highest. In contrast, the RF technique exhibited exceptionally favorable outcomes, achieving a magnificent score of 99%. These findings underscore the importance of innovative feature generation in attaining optimal results using machine learning.

The results illustrated in Fig. 12 showcase the efficacy of innovative feature extraction strategies in elevating the capabilities of machine learning methodologies. Radar chart analysis revealed that all employed techniques exhibited commendable performance, as

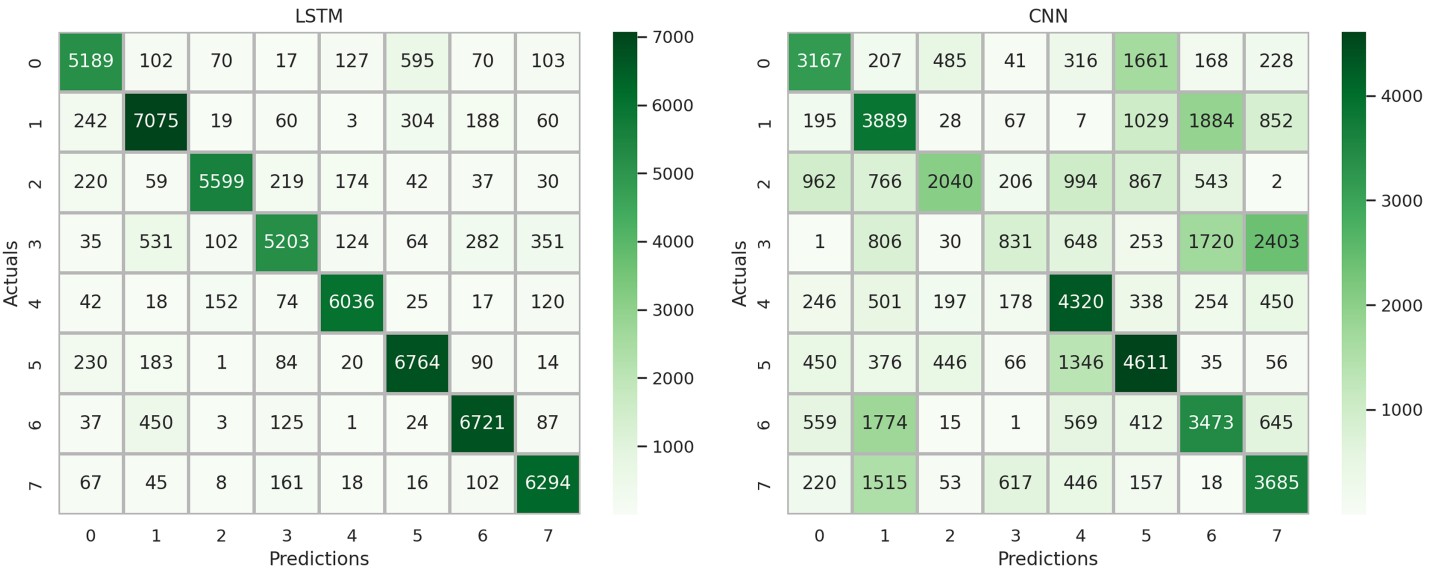

**Figure 10 Analysis of the confusion matrix outcomes for the employed deep learning methodologies using the original features.**

**Table 7 Performance metrics analysis results with proposed feature engineering with implemented machine learning models.**

| Technique | Accuracy | Target | Precision | Recall | F1 | Support |
|---|---|---|---|---|---|---|
| DT | 0.98 | Extended leg raises | 0.99 | 0.99 | 0.99 | 7,951 |
| | | Forward bending | 0.99 | 0.99 | 0.99 | 7,448 |
| | | Straight lying-leg raises | 0.98 | 0.98 | 0.98 | 7,386 |
| | | Side lying hip abduction | 0.99 | 0.99 | 0.99 | 6,484 |
| | | Alternating leg lifts prone | 0.97 | 0.98 | 0.97 | 6,273 |
| | | Elbow flexion | 0.99 | 0.99 | 0.99 | 6,711 |
| | | Shoulder abduction | 0.98 | 0.99 | 0.99 | 6,692 |
| | | Prone lying elbow extension | 1.00 | 1.00 | 1.00 | 6,380 |
| | | Average | 0.99 | 0.99 | 0.99 | 55,325 |
| LR | 0.97 | Extended leg raises | 0.96 | 0.97 | 0.97 | 7,951 |
| | | Forward bending | 0.97 | 0.95 | 0.96 | 7,448 |
| | | Straight lying-leg raises | 0.97 | 0.97 | 0.97 | 7,386 |
| | | Side lying hip abduction | 0.99 | 0.99 | 0.99 | 6,484 |
| | | Alternating leg lifts prone | 0.97 | 0.97 | 0.97 | 6,273 |
| | | Elbow flexion | 0.99 | 0.99 | 0.99 | 6,711 |
| | | Shoulder abduction | 0.96 | 0.96 | 0.96 | 6,692 |
| | | Prone lying elbow extension | 1.00 | 1.00 | 1.00 | 6,380 |
| | | Average | 0.98 | 0.98 | 0.98 | 55,325 |

(Continued)

| Table 7 (continued) | | | | | | |
|---|---|---|---|---|---|---|
| Technique | Accuracy | Target | Precision | Recall | F1 | Support |
| GNB | 0.96 | Extended leg raises | 0.95 | 0.94 | 0.94 | 7,951 |
| | | Forward bending | 0.97 | 0.94 | 0.95 | 7,448 |
| | | Straight lying-leg raises | 0.94 | 0.96 | 0.95 | 7,386 |
| | | Side lying hip abduction | 0.99 | 0.98 | 0.98 | 6,484 |
| | | Alternating leg lifts prone | 0.94 | 0.96 | 0.95 | 6,273 |
| | | Elbow flexion | 0.98 | 0.98 | 0.98 | 6,711 |
| | | Shoulder abduction | 0.93 | 0.95 | 0.94 | 6,692 |
| | | Prone lying elbow extension | 1.00 | 1.00 | 1.00 | 6,380 |
| | | Average | 0.96 | 0.96 | 0.96 | 55,325 |
| RF | 0.99 | Extended leg raises | 0.99 | 0.99 | 0.99 | 7,951 |
| | | Forward bending | 1.00 | 0.99 | 0.99 | 7,448 |
| | | Straight lying-leg raises | 0.99 | 0.99 | 0.99 | 7,386 |
| | | Side lying hip abduction | 0.99 | 1.00 | 0.99 | 6,484 |
| | | Alternating leg lifts prone | 0.98 | 0.98 | 0.98 | 6,273 |
| | | Elbow flexion | 1.00 | 1.00 | 1.00 | 6,711 |
| | | Shoulder abduction | 0.99 | 0.99 | 0.99 | 6,692 |
| | | Prone lying elbow extension | 1.00 | 1.00 | 1.00 | 6,380 |
| | | Average | 0.99 | 0.99 | 0.99 | 55,325 |

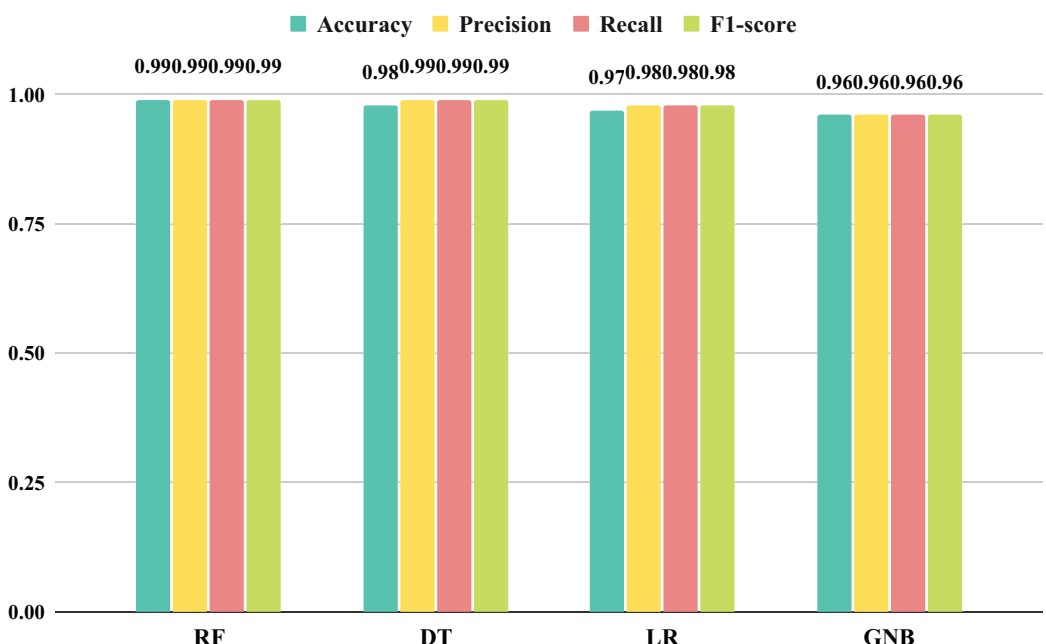

**Figure 11 The results analysis of performance for applied machine learning models with proposed transfer features.**

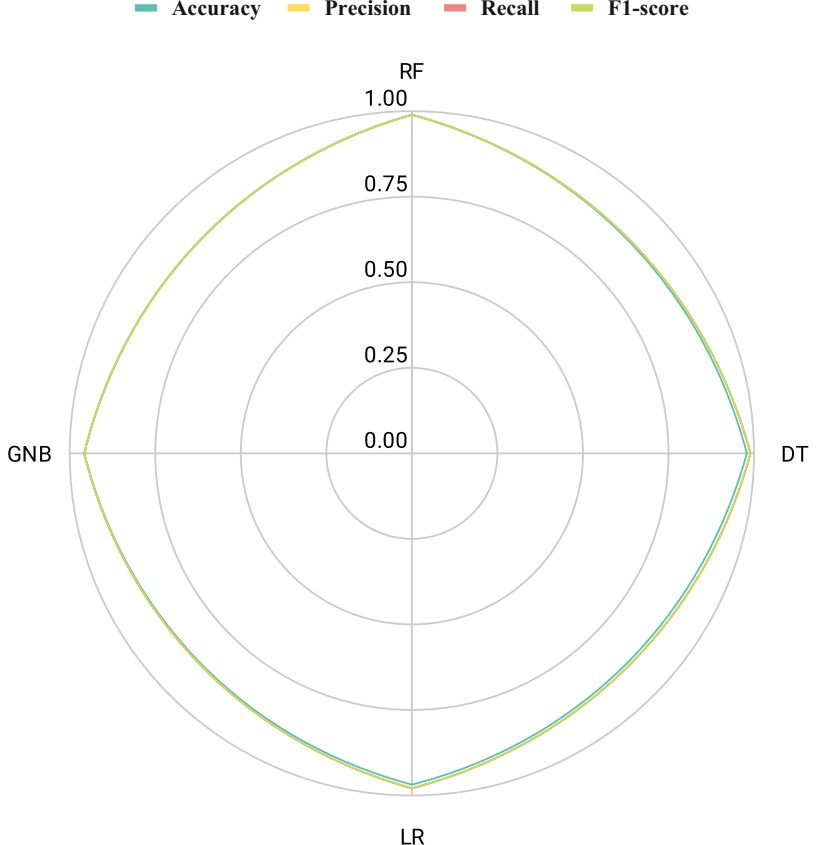

**Figure 12 An assessment of the performance results achieved by employed methods utilizing proposed transfer features through radar chart analysis.**

indicated by the extensive coverage of the performance metric curves within the radar boundaries. While DT, GNB, and LR techniques scored satisfactorily, they did not attain the highest levels. What is particularly noteworthy is the exceptional performance of the RF technique, which encompassed a substantial area in the radar graph and achieved maximal scores across all metrics. These outcomes imply that the proposed feature generation approach holds the potential to significantly enhance the efficiency of machine learning methods, with RF emerging as the most encouraging approach for optimal performance.

A comprehensive evaluation of the performance of different methods incorporating a novel feature extraction approach is presented in Fig. 13 through a detailed analysis of the confusion matrix. The findings from the analysis indicated a significant reduction in error rates for the target classes across all applied methods, demonstrating consistently high-performance scores. Notably, the RF method, as depicted in its confusion matrix, shows minimal error rates for the target classes, along with commendable accuracy scores across various classes. This analysis concludes that the proposed feature generation approach based on ensemble learning has the potential to significantly enhance the performance of machine learning models in the classification of physical therapy exercises.

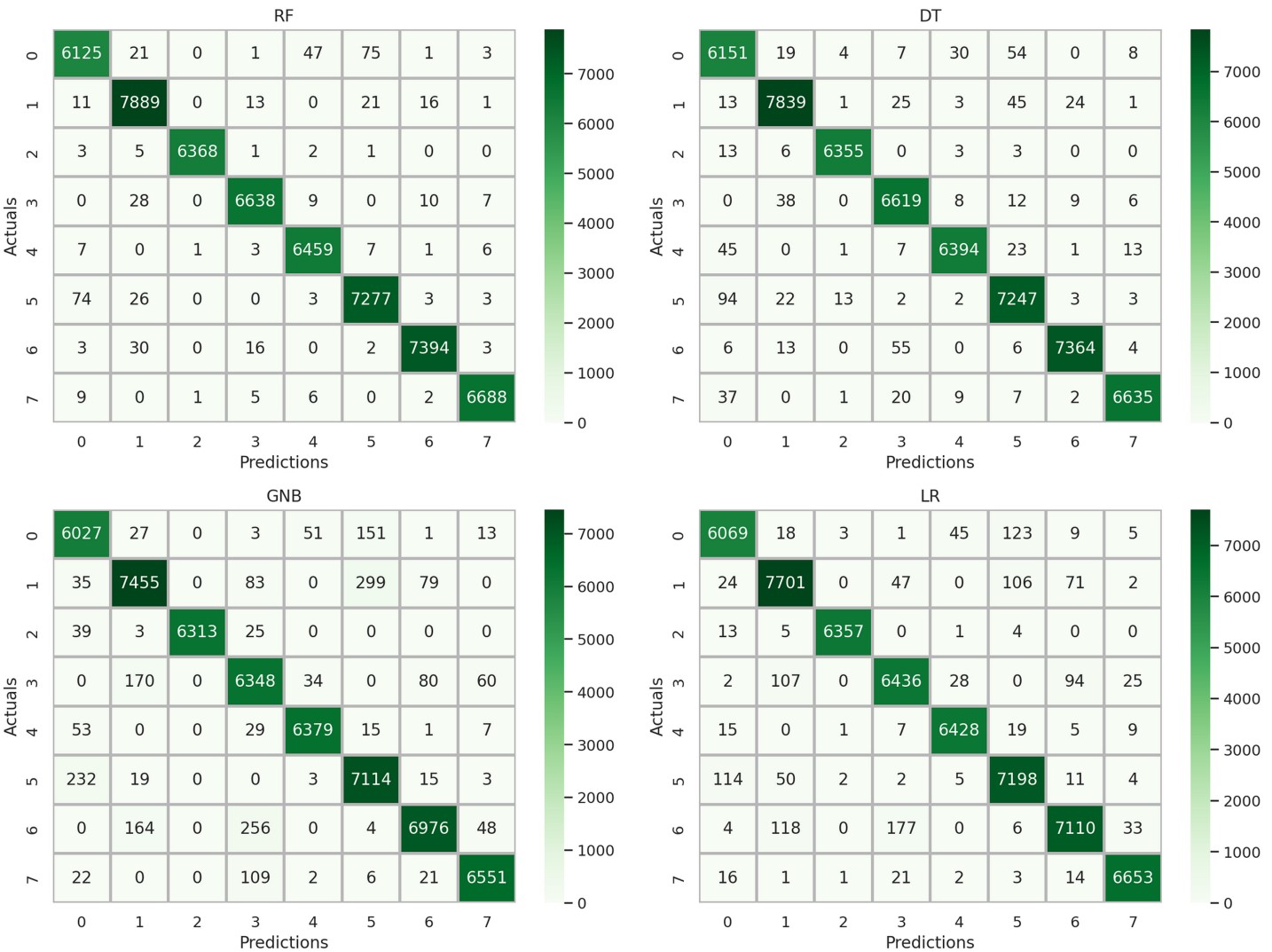

**Figure 13 Analysis of the results from the confusion matrix for the implemented techniques using the proposed transfer features.**

## K-fold-based performance validations

The efficacy of each approach, coupled with the proposed feature engineering, was thoroughly assessed through K-fold analysis (*Anam et al., 2021*), as detailed in Table 8. The results indicated that integrating a new feature dataset significantly enhanced the effectiveness of the implemented machine-learning methods. This improvement was evident in the increased K-fold cross-validation scores and reduced standard deviation observed. Among the techniques evaluated, RF emerged as the most effective, achieving exceptional K-fold accuracy of 0.99 and demonstrating the lowest standard deviation (0.005) compared to other methods. This underscores the success of the proposed RFL feature engineering approach in enhancing the generalizability of machine-learning algorithms for classifying physical therapy exercises. The findings highlight the potential of

**Table 8 K-fold cross-validation performance results analysis of applied method with proposed transfer feature engineering.**

| Technique | Fold | K-fold accuracy | Standard deviation (+/−) |
|---|---|---|---|
| DT | 10 | 0.98 | 0.0006 |
| LR | 10 | 0.97 | 0.0007 |
| GNB | 10 | 0.96 | 0.0009 |
| RF | 10 | 0.99 | 0.0005 |

**Table 9 Analysis of the computational complexity in applied methods utilizing the suggested transfer feature engineering approach.**

| Technique | Runtime computation (S) |
|---|---|
| RF | 6.879 |
| DT | 3.063 |
| LR | 17.32 |
| GNB | 0.347 |

this innovative feature extraction approach to improve the accuracy and reliability of classifying and correcting physical therapy exercises across various applications.

## Computational complexity analysis

An analysis of the computational complexities of various machine learning approaches, combined with the proposed feature engineering, is detailed in Table 9. This examination illustrates the enhanced efficiency of machine learning methods when utilizing the proposed novel feature extraction method compared to using original features alone. Specifically, the LR technique had the longest computational runtime score of 17.32 s, while the GNB method demonstrated the shortest runtime score of 0.34 s, albeit with lower performance accuracy. These findings indicate that the proposed RF technique, coupled with the novel feature engineering approach, achieved a reduced runtime computational score. In addition, the scalability of the RFL model is analyzed by evaluating its computational runtime with increasing dataset sizes. As the dataset size increased, the RFL model demonstrated a moderate rise in runtime, indicating a computational cost higher than traditional models.

## Comparison of feature space

A comparative analysis of feature space representations, depicted in Fig. 14, compares original features with a recently developed set. The examination suggests that original features from the wearable inertial and magnetic sensor dataset lack linear separability, resulting in sub-optimal performance with machine learning techniques. In contrast, our proposed feature engineering approach results in a newly created feature set that exhibits enhanced linear separability. Overall, increased linear separability in our proposed features contributes to superior performance in the classification of physical therapy exercises.

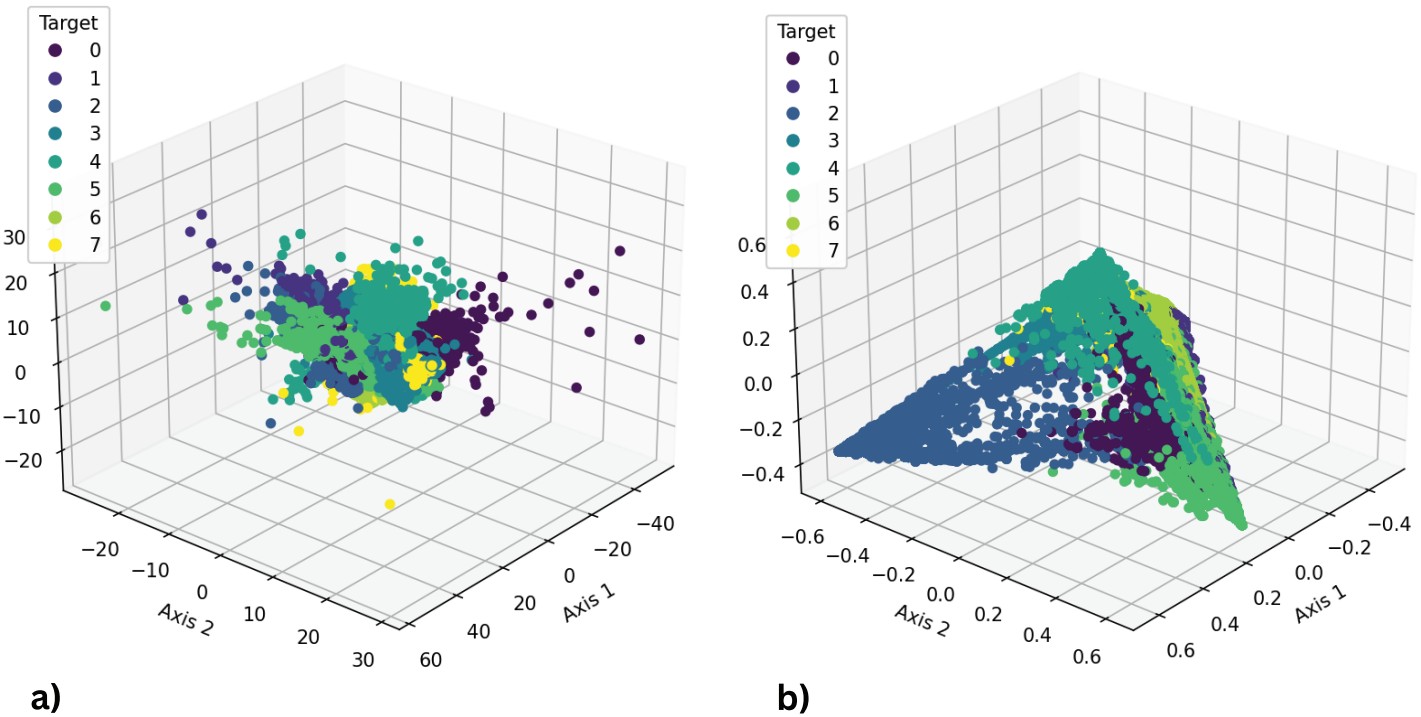

a)                                                                 b)

**Figure 14** (A) Representations of original features; (B) representations of the newly generated features.

## State of the art comparisons

Our study conducted a comparative performance analysis, detailed in Table 10, across studies published from 2020 to 2023 focusing on the classification of physical therapy exercises. These studies primarily utilized deep learning and machine learning models to achieve favorable scores. Our findings consistently show superior results compared to these studies, with our proposed approach achieving high accuracy scores in the classification of physical therapy exercises.

### Results with independent data

To assess the robustness and generalizability of the RFL method, we evaluated the model's performance on an independent test dataset *Fedesoriano (2024)*. The results demonstrate a high accuracy of 99.45%, with an error rate of only 0.54%, indicating that the model effectively distinguishes between classes. The precision, recall and F1-score values are consistently high, further confirming the method's reliability, as shown in Table 11.

## Discussions and limitations

In this study, an innovative approach to feature engineering was employed to classify physical therapy exercises with exceptional efficiency. Various machine learning and deep learning techniques were implemented and compared to assess their performance. Each method's performance was rigorously validated through the k-fold technique and hyperparameter training. The extensive analysis of results demonstrates that the utilization

**Table 10 The performance comparisons of our novel proposed method with state of the art techniques.**

| Ref. | Year | Learning type | Proposed technique | Performance accuracy (%) |
|---|---|---|---|---|
| Uslu et al. (2020) | 2020 | Machine learning | SVM | 90 |
| Francisco & Rodrigues (2023) | 2022 | Machine learning | MNN | 90 |
| Carrera, Arequipa & Hernández (2022) | 2022 | Deep learning | RNN | 95 |
| Bansal & Vishwakarma (2023) | 2023 | Machine learning | KNN | 92 |
| Kanungo et al. (2023) | 2023 | Machine learning | ML system | 96 |
| Our | 2024 | Transfer learning | RFL | 99 |

**Table 11 Performance evaluation of RFL method with an independent dataset.**

| Metric | Value |
|---|---|
| Accuracy | 99.45% |
| Error rate | 0.54% |
| Training time | 0.0649 s |
| Precision (Avg.) | 1.00 |
| Recall (Avg.) | 0.99 |
| F1-score (Avg.) | 0.99 |

of the newly proposed feature engineering methods leads to impressive performance improvements. In contrast, the original dataset features exhibited lower performance scores, as confirmed by the feature space analysis. Furthermore, computational cost analysis highlighted the efficiency of the proposed method in this study.

In conclusion, our introduced approach has the potential to significantly transform the classification of physical therapy exercises by achieving high-performance scores. A comprehensive performance comparison with state-of-the-art approaches further establishes the superiority of our proposed technique in effectively classifying and enhancing physical therapy exercises.

The potential drawbacks and challenges encountered during the study:

- Sensor data variability: Differences in sensor placement, calibration, and user movement variations could introduce inconsistencies in the collected data, affecting model generalization.
- Feature selection challenges: Extracting meaningful transfer features from LSTM and RF models required careful tuning, and suboptimal selections could degrade model performance.
- Real-time implementation: While the proposed model demonstrated high accuracy, its feasibility for real-time applications, particularly in resource-constrained devices, remains a challenge.

The proposed model has the potential to significantly reduce the cost and frequency of clinic visits by providing real-time feedback on physical therapy exercises. By leveraging wearable sensor data, the model can accurately classify and correct exercises without

requiring constant supervision from physiotherapists. This enables patients to perform rehabilitation exercises at home with immediate guidance, minimizing the need for in-person consultations. Another key application is home-based physiotherapy, where patients can perform prescribed exercises independently while receiving real-time feedback from the model. This reduces reliance on frequent clinic visits, making rehabilitation more convenient and cost-effective.

## CONCLUSIONS

This study introduces a novel technique for classifying and correcting physical therapy exercises using advanced machine learning approaches to achieve high-performance results. The investigation utilized publicly available data from wearable inertial and magnetic sensors for experimentation. Four sophisticated machine-learning techniques were implemented and compared. A novel RFL method was proposed for feature generation from sensor data, which was then used to develop learning approaches for classifying physical therapy exercises. The study results indicate that the proposed RFL approach achieves the highest performance in correcting physical therapy exercises. Extensive experiments showed that employing RF within the proposed RFL method achieved a remarkable performance score of 99%. Performance was rigorously validated using the k-fold method and hyperparameter optimization. Additionally, comprehensive analyses of computational complexity and feature space confirmed the superior performance of the proposed approach.

### Future Directions

In future work, our plan includes developing a graphical user interface (GUI) integrating our proposed machine-learning model into the backend. The framework will receive real-time sensor data from patients during physical therapy exercises and provide corrections by alerting them if exercises are performed incorrectly.

### Funding

This research was supported by the MSIT (Ministry of Science and ICT), Korea, under the ITRC (Information Technology Research Center) support program (IITP-2024-RS-2024-00437191) supervised by the IITP (Institute for Information & Communications Technology Planning & Evaluation). The funders had no role in study design, data collection and analysis, decision to publish, or preparation of the manuscript.

### Grant Disclosures

The following grant information was disclosed by the authors:
MSIT (Ministry of Science and ICT), Korea.
ITRC (Information Technology Research Center) support program: IITP-2024-RS-2024-00437191.
IITP (Institute for Information & Communications Technology Planning & Evaluation).

## Competing Interests
The authors declare that they have no competing interests.

## Author Contributions
- Aisha Naseer conceived and designed the experiments, performed the experiments, analyzed the data, performed the computation work, prepared figures and/or tables, authored or reviewed drafts of the article, and approved the final draft.
- Ali Raza conceived and designed the experiments, performed the experiments, analyzed the data, performed the computation work, prepared figures and/or tables, authored or reviewed drafts of the article, and approved the final draft.
- Hadeeqa Afzal conceived and designed the experiments, performed the experiments, performed the computation work, prepared figures and/or tables, authored or reviewed drafts of the article, and approved the final draft.
- Aseel Smerat conceived and designed the experiments, performed the experiments, authored or reviewed drafts of the article, and approved the final draft.
- Norma Latif Fitriyani conceived and designed the experiments, performed the experiments, analyzed the data, performed the computation work, prepared figures and/or tables, authored or reviewed drafts of the article, and approved the final draft.
- Yeonghyeon Gu conceived and designed the experiments, analyzed the data, authored or reviewed drafts of the article, funding acquisition, supervision, and approved the final draft.
- Muhammad Syafrudin conceived and designed the experiments, analyzed the data, authored or reviewed drafts of the article, funding acquisition, supervision, and approved the final draft.

## Data Availability
The Physical Therapy Exercises dataset is available at UCI Machine Learning Repository: Yurtman, A. & Barshan, B. (2014). Physical Therapy Exercises [Dataset]. UCI Machine Learning Repository. https://doi.org/10.24432/C5JK60.

The code is available in the Supplemental Files.

## Supplemental Information
Supplemental information for this article can be found online at http://dx.doi.org/10.7717/peerj-cs.2854#supplemental-information.

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
