# Peer review of "Human pose estimation in physiotherapy fitness exercise correction using novel transfer learning approach"

_PeerJ Computer Science, doi:10.7717/peerj-cs.2854_

## Round 0.1 · original submission · Major Revisions

Dear Authors,

please read carefully the reviews and answer each point raised by the reviewers.

Best regards,

M.P.

Reviewer 1 ·

Basic reporting

1- The dataset features must be described in more detail, including total size, train-test split, and class distribution. These details should be organized into a table for clarity.
2- Authors should carefully proofread the manuscript to eliminate grammatical errors and typos to improve readability and professionalism.
3- Figures should be enhanced with higher resolution to improve their clarity and visual appeal for readers.

Experimental design

1. A flowchart and the detailed steps of the algorithm need to be included to provide a clear and concise understanding of the methodology.
2. The time spent on the experimental procedures, including model training and testing, must be measured and reported in the experimental results section.
3. A table summarizing the parameters used for the analysis (e.g., hyperparameters, batch sizes, optimizers) must be added for reproducibility.
4. The architecture of the proposed model should be described in greater detail to provide insights into its design and operational efficiency.

Validity of the findings

1. All performance metrics (e.g., accuracy, precision, recall, F1 score, AUC) should be calculated and included in the experimental results section for a comprehensive evaluation of the model.
2. A dedicated section on limitations and discussion should be added to address potential drawbacks and challenges encountered during the study.
3. The costs associated with deploying deep learning models, including necessary hardware and software, must be discussed to provide a practical perspective on implementation feasibility.

Additional comments

1. A section on potential future directions should be added to the conclusion, outlining how the research could be extended or improved.
2. To improve the "Related Work" and "Introduction" sections, the authors are recommended to review and reference the following related research papers:
a) Secure and Transparent Lung and Colon Cancer Classification Using Blockchain and Microsoft Azure.
b) Feature Reduction for Hepatocellular Carcinoma Prediction Using Machine Learning Algorithms.
c) A Novel Hybrid Approach to Masked Face Recognition Using Robust PCA and GOA Optimizer.
d) An Accurate System for Face Detection and Recognition.
e) Face Recognition Based on Grey Wolf Optimization for Feature Selection.
f) New Edge Detection Technique Based on Shannon Entropy in Gray Level Images.
g) A New System for Extracting and Detecting Skin Color Regions from PDF Documents.
h) Optimizing Epileptic Seizure Recognition Performance with Feature Scaling and Dropout Layers.
i) Innovative Hybrid Approach for Masked Face Recognition Using Pretrained Mask Detection and Segmentation, Robust PCA, and KNN Classifier.

·

Basic reporting

1. The text uses professional English but could improve in accessibility for non-specialists. Simplify highly technical sections, particularly in the Methodology.
2. The context and references are sufficient, but the knowledge gap and novelty could be better emphasized in the Introduction.
3. The manuscript mentions a public dataset but does not provide clear access instructions. Include a link to the dataset or repository.

Experimental design

1. Clearly defined and relevant. The novelty of integrating LSTM and Random Forest is compelling.
2. Detailed but requires better reproducibility. Include more pseudocode or diagrams for the RFL process.
3. Ethical considerations are not added. Acknowledge participant consent and dataset limitations.

Validity of the findings

1. Results are well-supported, but confidence intervals and external validation are necessary.
2. Test the RFL method on independent datasets to confirm its robustness and generalizability.

Additional comments

Dear Authors - Thank you for such a nice article, I enjoyed reading it, I believe it has a great potential to find its use in real world. Please refer for peer review comments.

Abstract and Introduction:
Informative and provides good context

1. Review the "Introduction" section (lines 34-68) for technical jargon and add brief explanations or references (e.g., “transfer learning,” “LSTM”).
2. Revise sentences like “The original smartphone wearable sensor data were processed via RFL to generate new temporal and probabilistic features” (Abstract). Simplify or break them into shorter sentences.
3. Add a sentence explicitly stating the novelty of the Random Forest Long Short-Term Memory (RFL) method and its broader implications in physiotherapy.
4. Include a paragraph highlighting limitations of prior physiotherapy exercise correction techniques, particularly CNN-based or Gaussian models.
5. Expand the novelty section by explaining how RFL uniquely integrates feature engineering to improve accuracy.

Methodology:
The methodology is comprehensive

1. Provide more details about the demographic diversity of participants and why the dataset is sufficient for physiotherapy exercises.
2. Clearly explain why the “time index” column was excluded, as mentioned in the Preprocessing subsection.
3. Could you add explicit formulas for feature extraction to improve reproducibility.

Results:
The results are well-organized.

1. Provide confidence intervals or standard deviations for the accuracy, precision, recall, and F1-score metrics to strengthen statistical validity (Tables 4–6).
2. Test the RFL method on an external, independent dataset to demonstrate generalizability.
3. Revisit the analysis of low-performing methods like GNB and LR to explain their shortcomings in physiotherapy contexts.

Computational Complexity:
Computational efficiency is mentioned briefly but could use deeper analysis

1. Expand the computational runtime analysis (Table 8) to compare the scalability of RFL when dataset sizes increase.
2. Discuss potential trade-offs between computational efficiency and accuracy for real-time use.

Practical Applications
The application potential of the RFL method is discussed but not elaborated.

1. In the Discussion, provide specific use cases such as home-based physiotherapy or integration into wearable devices.
2. Discuss how this model could reduce the cost and frequency of clinic visits by offering real-time feedback.

Conclusion
1. Add future directions such as exploring larger datasets or adapting the model for real-time feedback.

---

## Round 0.2 · accepted · Accept

All comments have been addressed by the authors; the manuscript is ready for publication.

Reviewer 1 ·

Basic reporting

-

Experimental design

-

Validity of the findings

-

Additional comments

-

·

Basic reporting

The version 2 looks good. Thanks for fixing comments.

Experimental design

The version 2 looks good. Thanks for fixing comments.

Validity of the findings

The version 2 looks good. Thanks for fixing comments.

Additional comments

The version 2 looks good. Thanks for fixing comments. I am accepting this article.